# Real-Time Flood Forecasting and Warning: A Comprehensive Approach toward HCI-Centric Mobile App Development

**Waleed Alsabhan \*** and **Basil Dudin**

Department of Software Engineering, Alfaisal University, Riyadh 11533, Saudi Arabia
\* Correspondence: walsabhan@alfaisal.edu

**Abstract:** This article discusses the design, development, and usability assessment of a mobile system for producing hydrological predictions and sending flood warnings in response to the desire for human-centered technology to better the management of flood occurrences. Our work acts as a bibliographic reference for understanding what others have attempted and found, as well as gives an integrated set of recommendations. Furthermore, our guidelines offer guidance to aid in the design of mobile GIS-based hydrological models for mobile devices. We concentrate on the full design of a human–computer interaction framework for an effective flood prediction and warning system. In addition, we analyze and address the current user needs and requirements for building a user interface for mobile real-time flood forecasting in a methodical manner. Although a functional prototype was created, the primary objective of this research was to comprehend the complexity of possible users' demands and actual use situations in order to solve the problem of comparable systems being difficult to use. After consulting with possible consumers, application design standards were established and implemented in the initial prototype. Focusing on user demands and attitudes, special consideration was given to the usability of the mobile interface. To develop the application, a variety of assessment methods are added. The conclusion of the examination was that the system is efficient and effective.

**Keywords:** human–computer interaction; HCI; usability studies; mobile app; flood forecasting; real-time flood forecasting and warning

## 1. Introduction

Floods are one of the natural dangers that pose a direct threat to the civilian population, routinely inflicting severe property damage and claiming a large number of lives. In recent decades, flooding has caused far more damage. There is a considerable likelihood that this trend will continue, and that the severity of floods will continue to grow [1]. This is mostly due to the rise in localized, short-term flood occurrences. Through forecasting and early warning systems, disaster relief personnel and the affected population must be reliably and promptly alerted to impending dangers in order to mitigate the destructive effects of flooding and to take targeted preventive measures.

Models of hydrological and hydraulic forecasting serve as the foundation for disaster relief decision making. However, model-based flood forecasts are often unclear and prone to inaccuracy. This is due to model and precipitation prediction uncertainty, as well as insufficient temporal and geographical hydrological input factors. Predictions are particularly challenging for smaller bodies of water because of the very rapid reactions of the basin, providing little time for warning. Frequently, there have been inadequate official statistics for these rare circumstances. However, new information, such as more water level measurements along a river, may be utilized to expand these essential datasets and update the prediction models in order to minimize forecasting uncertainty. Through mobile crowdsourcing and sensing [2], such extra hydrological data may also be freely captured and shared by people using the sensors in their own devices (e.g., cellphones and tablets).

These location-specific data are often referred to as volunteered geographic information (VGI) data [2]. By adding VGI data, the quantity of geographical and temporal input data for prediction models may be enhanced, thus lowering uncertainty and enhancing the flood forecasting systems' predictions. Involving the people in the process also increases the awareness of flood risks, and individuals may actively contribute to improving flood predictions and minimizing flood damage [3].

The purpose of this research is to examine if human–computer interaction (HCI) may enhance the usability of geographic information systems (GISs) and hydrological modeling on mobile devices. The paper presents the development of a real-time flood forecasting and warning system which combines a geographic information system (GIS) with dynamic hydrological modeling, with a focus on the user experience side of the end-product. The key aim of the work was to examine the design and evaluation process, which started with gathering a very broad set of requirements based on users' needs and technical possibilities, and then proceeding to create a user-friendly prototype on the basis of those requirements. The process was applied to an area where the user interfaces are typically difficult to use and involved groups of users with diverse levels of knowledge, technical capabilities, and backgrounds. The entire work deals with three core research questions:

RQ1. What framework can be used to comprehensively design a logical data-gathering workflow for effective flood prediction and warning systems in terms of human–computer interaction?

RQ2. How can we systematically investigate and approach the current user needs and requirements pertaining to designing a user interface for mobile real-time flood forecasting?

RQ3. How can we evaluate user feedback in light of the various app development stages to better comprehend human–computer interaction?

The remainder of the paper is organized as follows. In Section 2, related work, along with the core contributions of this work, is discussed. Section 3 presents the proposed framework answering RQ1. Section 4 introduces the user needs and requirements, answering RQ2. A usability evaluation with the app prototype is presented in Section 5, answering RQ 3. Lastly, a conclusion with future recommendations is presented in Section 6.

## 2. Related Work

Human–computer interaction research is essential for enhancing the performance of computer systems [1]. The science of human–computer interaction (HCI) must address new problems as individuals are increasingly seen as active agents and not just as collections of properties of cognitive processors. As the actual usage of computer systems over time becomes a concern and the objective of design switches from building single-user systems to designing systems that allow groups of people to collaborate, it is vital to concentrate on human–computer interaction. HCI practitioners sense the need for tools that will allow them to study the interaction that the systems they create support and organize. In summary, there is a growing consensus that a deeper understanding of context is necessary. Activity theory might potentially fill this hole. Activity theory is a framework that may assist designers and researchers in asking the proper questions to address complicated issues, but it does not provide a readymade answer [4]. This is in contrast to conventional theories that serve as prediction models. Activity theory resembles metatheory more than predictive theory [4]. Activity theory has been applied to HCI by many researchers in many fields [5–8]. Subject–object interaction is a notion from activity theory that resembles human–computer interaction. However, it is difficult to apply this approach to comprehend how individuals utilize interactive technology. Computers are often not objects of activity but rather artifacts that facilitate interaction. In other words, human interaction with the world is mediated by computers. The hierarchical organization of human activity is another idea. Frequently, the usage of a computer (or other technology) occurs at the operational layer, and it is required to tie such operational features to higher levels, such as significant objectives and the requirements and motivations of technology users. Similarly, in scenario-based design, the usage of a future system is outlined in detail early in the

development process [9]. Then, narrative descriptions of anticipated use episodes are used in a number of ways to influence the construction of the system that would facilitate these usage experiences. Similar to other user-centered techniques, scenario-based design shifts the emphasis of design work from specifying system functions to describing how individuals would use a system to complete work tasks and other activities. In contrast to methods that examine human behavior and experience via formal analysis and modeling of well-defined tasks, scenario-based design is a very lightweight tool for imagining future usage possibilities. An interaction scenario with the user is a sketch of usage [10]. In the same way that a two-dimensional, paper-and-pencil drawing captures the essence of a physical design, interaction design wireframes are meant to vividly represent the essence of an interaction design [11]. There have been many cases of using these approaches in terms of human–computer interaction and flood prediction and warning systems [12–16]. The use of different technologies in order to accomplish this is another important task. Several working factors are important here.

Firstly, improving user–machine interactions for flood prediction and warning systems is difficult because flood information systems must operate in real time to facilitate coordination among flood agencies, organizations, and affected citizens [3], and because predictive environmental sensor networks cover large geographical regions of interest and support multiple sensor types to detect relevant phenomena [17]. Moreover, GIS user interfaces are often developed without consideration for accessibility or the specific demands of crisis managers required to make choices under stressful emergency settings [1]. Although collaborative decisions based on geographic data are essential to the management of emergency situations, GISs are not built to handle several users concurrently [18]. In recent years, Nivala et al. [19] documented several initiatives to enhance the user experience of flood prediction and warning systems, with an emphasis on GIS. GIS user interfaces are often difficult to use, which can reduce the effectiveness of emergency management. Kadlec et al. argued for the importance of usability studies for improving them and proposed HydroDesktop, an open-source GIS application with features designed to improve the user experience: spatial and temporal filtering and interpolation, simultaneous graph and map display, and spatial features linked to time-aggregated observation data [20].

Large interactive surfaces have been shown to improve collaborative decision making. Döweling et al. proposed an interactive tabletop system to improve collaborative situation analysis and planning [21]. Döweling et al. showed its effectiveness in a contrasting study where 30 participants were tasked with improving the efficacy of crisis management through utilizing the tabletop system, traditional paper maps, and a basic desktop GIS program. The tabletop system made users more efficient, and they considered it a superior user experience conducive to teamwork. Resch and Zimmer analyzed the application potential for a "map-based geo-portal user-experience perspective" to address the incompatibility between different approaches to design and usability [22]. They concluded that standardizing "the user-experience design of map-based geo-portals" is an important way of improving their effectiveness.

The importance of human interaction in effective flood prediction and warning systems has also been shown by Kushwaha et al. [23] and Mosavi et al. [24], who conducted a usability analysis of weather forecast data sent from the National Center for Medium-Range Weather Forecasting (NCMRWF) in Noida, India, to the agro-meteorological field unit Pantnagar. Feedback from Pantnagar to NCMRWF helped produce accurate weather forecasts, equipping farmers to improve their production. Increased usability could further improve coordination between the two units and, hence, farmers' ability to make informed decisions. The new version of the real-time Global Flood Monitoring System (GFMS), powered by multi-satellite rainfall analysis of the Tropical Rainfall Measuring Mission (TRMM), is an important innovation in flood prevention. It supports larger affected areas and longer flood durations [25].

An important consideration is whether to use a mobile app or a dashboard user interface. There are multiple advantages of using a flood warning mobile app over a

dashboard user interface (tablet). A mobile app is accessible from anywhere and at any time, but a dashboard interface is restricted to the tablet's location. This allows users to get notifications and updates even while they are not in front of their tablet. Additionally, mobile applications are meant to be accessible and user-friendly, with intuitive interfaces and straightforward navigation. This makes it simpler for people to get critical information, such as flood alerts and evacuation routes, during an emergency. Moreover, mobile applications may be tailored to match the unique requirements of each user. For instance, depending on their location and risk level, users may create customized notifications for particular flood zones. Typically, dashboard interfaces do not provide this amount of flexibility. Furthermore, mobile applications can deliver real-time updates on flood warnings and weather conditions, which is essential in emergency circumstances. During a crisis, dashboard interfaces that need manual updates or do not deliver real-time information might be disadvantageous. Another important advantage is push notifications. Mobile applications may give users push notifications, which can be more effective than dashboard alerts. Users may be alerted to critical conditions, such as flash floods or evacuations, by the use of quick, attention-getting push alerts. Hence, in comparison to tablet dashboard user interfaces, mobile applications for flood warning offer various benefits, including mobility, accessibility, customization, real-time updates, and push alerts. These benefits make mobile applications a more efficient tool for obtaining flood alerts and reacting to emergency situations.

There are different types of flood warning mobile apps with various pros and cons. A summary is provided in Table 1.

**Table 1.** Pros and cons of different types of flood apps.

| Types of Flood Apps | Parameters to Consider | Pros | Cons |
| --- | --- | --- | --- |
| Government apps | • Coverage area<br>• Language availability<br>• Customization options | • Free to download and use<br>• Accurate and up-to-date flood warnings<br>• Additional features such as flood maps, evacuation routes, and emergency contacts | • Only cover a specific geographic area or region<br>• May not be available in all languages<br>• May have limited customization options |
| Private apps | • Customization options<br>• Cost<br>• Reliability and accuracy | • More customization options<br>• Cover a wider geographic area<br>• Additional features such as flood history and predictions | • May require a subscription or fee to access all features<br>• May not be as reliable or accurate as government apps<br>• May prioritize profit over public safety |
| Crowdsourced apps | • User feedback and reviews<br>• Frequency and quality of updates<br>• Reliability and accuracy | • Allow users to contribute and share flood information in real time<br>• May cover areas not covered by government or private apps<br>• Can be useful for quickly identifying flood-prone areas | • May not be as reliable or accurate as government or private apps<br>• May be vulnerable to false reports or misinformation<br>• May not have the same level of resources and support as government or private apps |

Our aim in this research is to develop a framework and app that can address the shortcomings of existing apps, as shown in Table 1.

Although significant work has been performed in these sectors, there is still a considerable gap in the proper approach framework, data gathering, and evaluation processes especially pertaining to flood prediction and warning systems. This article introduces and unique angle of utilizing the different frameworks for optimum outcomes. The research also suggests that risk management can be made more effective by incorporating human–computer interaction principles into the design of flood prediction and warning

systems. Our aim in this research is to develop a framework and app that can address the shortcomings of existing apps, as shown in Table 1. Additionally, we provide user feedback and evaluation. Real-time forecasting systems and improving the usability of the GIS interface are important steps in this process, as highlighted in the study with design principles, actual prototype development, and user feedback.

## 3. Framework Approach (RQ1)

In Section 2, we discussed the different approaches toward understanding user needs through human–computer interaction. The activity theory was chosen as a framework for understanding the complexity of potential users' needs to guide the development of the application for this work. The key advantage of the theory is understanding the full context of the use of an application (which helps in making an application accepted by users) which is critical in this situation because comprehension and acceptance of the product by end-users are strongly correlated with security and life-saving measures. An example of a situation where an existing system did not work to its full potential was the Mauritius Meteorological Service. The service was able to reach 10% of people before the tsunami, and only 42% of citizens knew about the warning system after the disaster. In addition, 64% of people continued their everyday life, and 15% did the opposite of what was recommended [26].

Clemmensen et al. showed that activity theory is well adapted for both social issues and the design of computer-based products [27], which is exactly what is needed for a flood forecasting application. Therefore, the theory was chosen as a framework, and its potential for creating such applications was further examined by this research project. The theory deals with the activity itself (driven by its motive), the action required for an activity, and the actual operations and conditions required to achieve a specific goal [21–24]. As shown in Figure 1, the concept contains the following elements: subject (an actor engaged in an activity), object (the intention at which an activity is directed), tools (used by the subject to achieve the object), rules (guidelines, conventions, and norms applied during an activity), division of labor (a division of activities among actors), and community (other actors and the social environment) [25]. When an activity takes place, relationships are formed and developed among all these elements. The main characteristics of activity theory are object-oriented, a hierarchical structure of activity, internalization and externalization, mediation, and development. Therefore, activity theory supports the description of complex systems with composite relations. The literature review highlighted the potential of activity theory for understanding users' needs: it helps in understanding the tasks that people are engaged in as part of an activity, their goals and how they achieve them, the reasons behind specific tasks, the meaning of the tasks for them, interactions with other people to complete their tasks, the roles of society and people's immediate environment in their performance, and lastly, what tools or instruments people use to attain their goals. Such complex understanding is essential for designing a novel system and ensuring its usability. Furthermore, the scenario-based design (SBD) approach was used as a tool for applying this framework. Scenarios describe the anticipated use of the system and allow designers to understand user interactions and needs [28,29]. A scenario describes a typical usage situation and mentions the actors involved and the relationships between them, as well as their goals, actions, and objects. Analyzing scenarios allows the expanded activity theory diagram to be filled in since the elements of activity are either covered in a scenario or can be gathered by discussing it. Brainstorming was used to generate scenarios since it is a common method and has been adopted by many researchers in the information design field [30–36] and beyond. Since research has proven that individual brainstorming generates more ideas than group brainstorming [37,38], the nominal group technique, proposed by Delbecq and Van de Ven [39] and illustrated procedurally by Sample [40], was used.

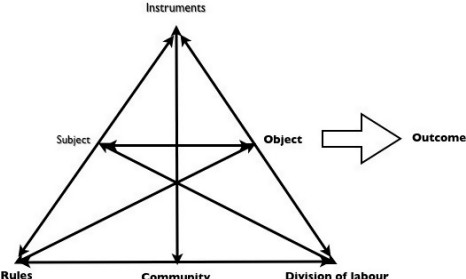

**Figure 1.** Engeström's expanded model of an activity system.

On the basis of the discussions above, we developed a framework for a real-time flood prediction and warning system through a mobile device. In terms of the capabilities of the system, it should be practical and applicable to a wide variety of watershed scenarios. It should be ensured that critical data about rainfall can be uploaded and are available to view and analyze in real time [23,24]. The paper focuses on users and their tasks as early as possible in order to understand the users' needs, as well as their expectations and mental models. In this regard, knowing the procedure of data collection and system components is crucial.

### 3.1. Data Collection

Before starting a discussion of the design, development, and evaluation of the interface used to produce the interface guidelines, we need to briefly explain the architecture of the flood prediction and warning system. Data capture imposes both hardware and software requirements. It requires specialized manual mechanisms such as automated data loggers (for collecting climate and hydrologic data), as well as rain gauges and specialized field computers for inputting data and writing to the database. Servers are also needed to store the data, and wireless modems to transmit the data inputted from the field and onto a storage server. The collected data are downloaded to a database simply by utilizing File Transfer Protocol (FTP), capturing them in almost real time, which the GIS model can immediately access and analyze. The software is required to convert data to compatible structures for the database architecture and to validate data before they enter the system. The data collection configuration is shown in Figure 2.

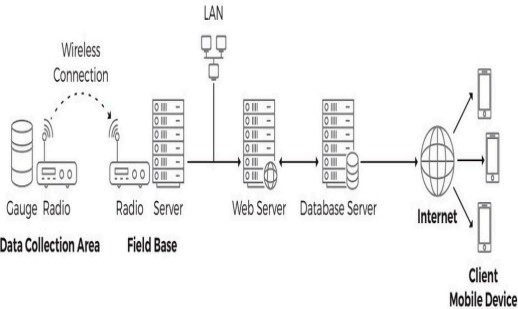

**Figure 2.** Project architecture.

### 3.2. System Components

The system comprises the following components: hydrological model (written in Java), ArcGIS, server-side PHP scripts (to manage data from a mobile handset), client side (written in Java; mobile), MySQL database, and a graphic tool. Most GISs do not easily support dynamic models because dynamic models were intended for querying and maintaining a static database. They do not explicitly support storage or analysis of dynamic phenomena or efficient iteration over time [41]. Therefore, integrating a dynamic model with existing systems requires the capability to receive and continuously process data and transfer the output data to the existing database.

The system incorporates a unique hydrological model incorporating a quantitative description and understanding of the processes. The details of this model are outside the scope of this paper. A similar model has already been developed and was used within an "offline spatial decision-support system for the MODULUS project" [42,43]. The model was selected because it utilizes a simple data-lean system, which has already been applied on a regional scale for hydrological purposes. Many other hydrological models were available, but none had the same simplicity of purpose and knowledge acquired through the previous application. It is also available in house and can, therefore, be accessed at a minimal cost. It provides accessible resources to support the project.

The dynamic modeling unit runs as a background process that applies hydrological analysis to the incoming data and returns the results, which can be spatial or nonspatial variables, such as rainfall or soil moisture.

*3.3. Data Forecasting Method*

Logistic regression was used to predict the likelihood of inundation on the basis of a number of environmental and climatic variables. Here is the used model for flood prediction using logistic regression:

1. Define the dependent variable: The dependent variable is the occurrence of inundation at a particular location during a specified time interval. This can be represented as a binary variable, with 1 representing inundation and 0 representing no flooding.
2. Identify independent variables: Environmental and climatic factors that may contribute to inundation are independent variables. Intensity of precipitation, river flow rate, soil moisture content, topography, land use, and season are some examples.
3. Collect data: Data were collected on the dependent and independent variables for a set of historical flooding events. The logistic regression model was trained using these data.
4. Divide the data: The data were divided into two sets: training and assessment. The training set was used to train the logistic regression model, while the testing set was used to assess the efficacy of the model.
5. Conform to the type: A logistic regression model was fit using maximum likelihood estimation to the training data. The form of the model was as follows:

$$P(\text{flooding}) = 1/(1 + \exp(-(b_0 + b_1x_1 + b_2x_2 + \ldots + b_nx_n))), \tag{1}$$

where $b_0$ is the intercept or bias term, and $b_1, b_2, \ldots, b_n$ are the coefficients for the independent variables $x_1, x_2, \ldots, x_n$.
6. Then, utilizing the testing set, the efficacy of the logistic regression model was determined. This was accomplished by calculating the model's precision, sensitivity, and specificity. Specificity is the proportion of true negatives, while sensitivity is the proportion of true positives (floods correctly predicted; no floods correctly predicted).

## 4. Systematic Investigation and Methodology (RQ2)

This section discusses the logical sequence of steps taken in order to build flood prediction and warning applications. We discuss the reasonings and justify our methodology. Furthermore, we introduce different design principles derived from the systematic investigation.

*4.1. Research Process*

The objective of the first stage of the research process was to gain valuable insights into (a) how potential users would interact with such applications, (b) what technical constraints should be considered during application design, and (c) how to transform these insights into actionable starting points for developing and testing the application. To achieve these objectives, we used professional reviews. Participating in the research were five possible (representative) users and five experts (a software developer, a software designer, a business analyst, and two GIS specialists). Participants varied in demographics (occupation,

age, and gender) and experience with mobile applications and weather-oriented systems (excluding those who are completely unfamiliar), including individuals with relevant expertise; GIS experts and software developers could comment on the technical viability of proposed scenarios, a designer could offer design-related insights, etc. After being randomly separated into two groups (using the nominal group approach), participants were asked to generate use cases for the proposed mobile real-time flood forecasting and warning system according to their experiences and requirements. The ideas were conveyed as written scenarios and documented on a whiteboard, and group members could ask for clarification without critiquing ideas, which stimulated debate and the development of new situations. The groups and the situations were then pooled for scenario assessment. A total of 63 situations were found. Twenty-two of the potential scenarios were immediately rejected as being beyond the scope of the research. Each participant anonymously scored the situations on a 10-point scale depending on how significant and worthwhile they deemed each scenario to be. On the basis of these point totals, 12 scenarios (those with at least 50% of available points) were chosen. Table 2 outlines the situations that were chosen. However, such circumstances did not occur. To complete the activity theory triangle, the participants and two user experience researchers dissected the highest-rated scenarios into instruments, subjects, objects, rules, communities, and divisions of labor, and then constructed the expanded activity theory graphic. The research team then studied the activity theory diagram and extracted needs from it. From the study and discussions, a list of design principles (generic guidelines and considerations) and design criteria (more detailed needs) was compiled.

**Table 2.** Selected scenarios.

| Rank | Scenario | Points |
|---|---|---|
| 1 | The city authority is notified by the local weather station that there is an increased likelihood of flood in the next 2 days. Combining statistical data about precipitation and terrain elevation models, the system informs the weather station and the local authority about the risk of a likely upcoming flood. | 105 |
| 2 | Martin is one of the officers in weather station X, which is connected with the system infrastructure; he noticed a change in the precipitation and moisture levels, and uses the system to update the current values. | 98 |
| 3 | Alice has just moved to a town, and she is quite curious to find out what the weather conditions (e.g., rainfall) are in this town. Thus, she uses the system to view historical data of such characteristics. | 94 |
| 4 | Bob wants to travel to London this weekend, but he does not know the weather, the likelihood of rainfall, and other weather parameters; thus, he uses the application to get the desired information. | 93 |
| 5 | Since the system provides information about rainfall, precipitation, moisture, etc., it would be extremely helpful for Bob, a farmer, to help him pick the best dates to plant his vegetables. | 87 |
| 6 | Bob uses the application in order to get notified when the likelihood of flooding in the various places he has entered is high, in order to take the actions required to address the threat and safeguard his properties. A quick guide on how to address the threat of flooding would be extremely beneficial. | 78 |
| 7 | The stored data have been lost or are inaccurate; hence, the system administrator enters the system to manage, check, and restore the data (e.g., pluviometric data and digital elevation models of the monitored areas). | 73 |
| 8 | Alice uses the user-friendly web interface of the application to upload some data about the location she currently lives, and keep the community aware of the new facts. However, Martin, the weather station officer, should first check and then activate the given data. | 69 |
| 9 | The system has been out of service, but the administration team works on the network, database, and security of the system to put it back online in a few minutes, ensuring the proper functionality of the system. | 68 |
| 10 | The local authority and the local weather stations record the floods of the previous years and evaluate the risky periods and locations. They produce a guide informing the citizens about the threat of flooding and what they should do to be better prepared. | 64 |
| 11 | In order to find the best place and date frame to cultivate his favorite fruit, Bob uses the application to compare different locations at different periods of time. | 64 |
| 12 | The textual information is too chaotic/confusing for Alice. She would definitely prefer to visualize the data in charts or graphs. | 61 |

*4.2. Human–Computer Interaction for Purpose-Driven App Development*

The main subjects are, as expected, front-end users wanting to view a range of weather data, weather-station users, and administrators who update and maintain the data. Requirements emerged for roles and access and modification rights, as well as a general requirement to provide different user interfaces for the roles.

As for objects, objectives for each role were identified, including updating the data accurately and informing authorities about upcoming severe weather events for weather-station users, getting information about pluviometric data and flooding conditions (with more specific objectives such as viewing historical data and performing a rainfall prediction) for front-end users, and maintaining the system and ensuring data integrity for administrators. Corresponding design requirements were listed.

In terms of community, administrators are members of the administrative team, which communicates with the weather-station users. Each user of a local weather station is a member of the local station community and of the national station community, which supports all the local weather stations. The front-end users, who represent the majority of the system's users, share information and experience with each other, with the weather-station users, and with people who do not currently use the system. They might also update the system with new data or notify the system moderators about an error state. This establishes requirements for effective collaboration between the groups, as well as requirements for adding users, getting notifications about new data being added to the system, etc.

Regarding the instruments, various tools and resources are required to accomplish the objectives. Firstly, a user panel is needed, presenting multiple settings and configuration tools to the users depending on their role, e.g., global display and network settings and various role-specific settings such as data handling tools for reviewing incoming information for weather-station users. Front-end users need various tools for viewing the information (e.g., pluviometry queries to answer questions such as "When was the last flood event in Bristol?"), meaningful presentation of spatial data (e.g., geographic characteristics and features of the selected location), ways of viewing and comparing historical data and making basic predictions, and data visualization (text, diagrams, icons, and maps) to help users understand the information. All this needs to be provided in a user-friendly way (see Figure 3).

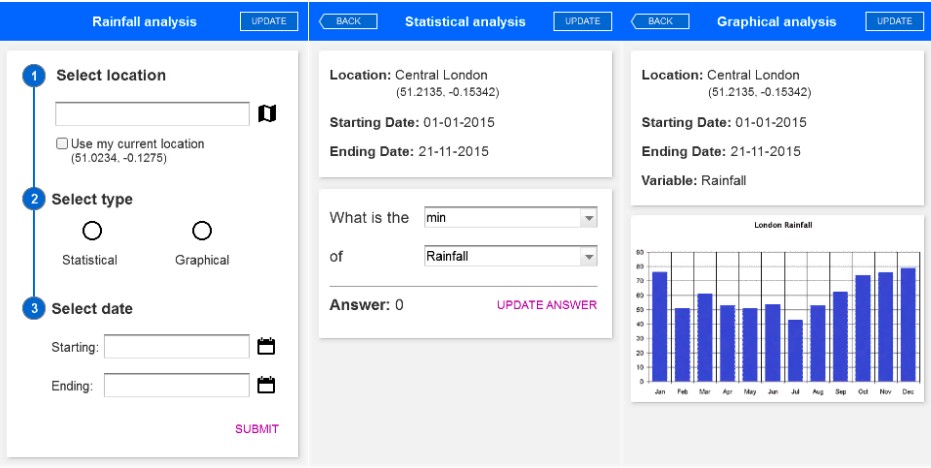

**Figure 3.** Some of the designed prototype screens for front-end users.

As for the division of labor, various roles are essential: maintenance personnel and physical and digital security personnel for the server where the data are stored, audit experts, system log analysts, network administrators, database administrators, data entry personnel, and personnel responsible for maintaining data collection devices. Internet service providers and energy suppliers are also important for the smooth running of the

application. Additionally, hydrology and GIS experts are needed to create a help section to show users how to conduct data analysis. All this stipulates requirements related to accurate data storage and transmission among the system's users, as well as integrity, maintenance, and security actions, and implementation of support mechanisms.

When it comes to rules, weather-station users will have to follow specific rules and procedures to keep the system up to date and notify the administrator about the changes. The administrators should also follow the rules and procedures to maintain the system and ensure the integrity of the data and the efficiency and performance of the system. Some social and other norms should be followed when users seek additional information or help to contact other people or web services. The corresponding requirements include that rules and regulations should be thoroughly described in a technical report document, and that the system should provide the system administrators with tools to define the rules for performing a search.

### 4.3. Design Principles

The investigated outcome was turned into a list of design principles that should guide the design and functionality of the application. The nature of the identified design principles is different from well-known heuristic guidelines from Nielsen, Norman, Baker, etc., which are well referenced in HCI research because our design principles are specific to real-time weather and flood warning systems instead of being general usability heuristics. Furthermore, they also focus on functionality instead of focusing mainly on usability. The design principles are presented in Table 3.

**Table 3.** Design principles.

| | |
|---|---|
| DP1 | The users should be divided into groups with discrete rights and responsibilities, depending on their roles. The three main user groups are the front-end users, the weather-station users, and the administrators. |
| DP2 | The system should provide information about pluviometric data and DEMs of the various monitored areas. |
| DP3 | The weather-station users should be able to manage data periodically and effortlessly, either automatically or manually, over their network channel. They should also be able to communicate with local authorities easily and quickly. |
| DP4 | Administrators should be able to maintain the system and ensure the integrity of the data and the efficiency and performance of the system. |
| DP5 | The front-end users should be able to provide and get information about pluviometric data through sophisticated filtering mechanisms. |
| DP6 | The system should provide mechanisms to support the communication between the different user groups and collaboration within these groups. |
| DP7 | A user panel should be used to facilitate user objectives, supporting different levels of abstraction and taking into consideration the different user roles and rights. |
| DP8 | The system should provide its users with meaningful pluviometric data of high detail after a request is performed, which should support sophisticated filtering and advanced search features. |
| DP9 | The system should provide spatial and nonspatial data with statistical pluviometric data. |
| DP10 | The users should be able to view historical data and filter them under various conditions, also getting information about the available samples. |
| DP11 | Data should be properly displayed and visualized, when needed, with the use of diagrammatic conventions and icons, along with textual information, ensuring usability and accessibility. |
| DP12 | The system data should be stored and transmitted accurately and with integrity among the system users, implementing supportive mechanisms. |
| DP13 | A set of rules and relegations should be applied to the system, to ensure its usability, functionality, operability, accessibility, and security. This set is defined by technological and user-experience factors. |

### 4.4. Design Requirements

On top of the design principles, we also identified 87 design requirements (Table 4) which aimed to provide more precise starting points for designing the prototype. Both

design principles and design requirements emerged from analyzing the scenarios, detailed discussions, and team members' expertise; the difference between the two is that design principles are broad enough to be reusable and guiding, while design requirements are more specific and actionable. An example for the first design principle is as follows: "The users should be divided into groups with discrete rights and responsibilities, depending on their roles. The three main user groups are the front-end users, the weather-station users, and the administrators". Examples of the design requirements are as follows: "The system supports three user roles: front-end users, weather-station users, and administrators." "Each role has discrete access/view/write rights in the system." "For different roles, different user interface layouts are supported." "A log-in and log-out mechanism should be supported." Design requirements focus on the actual functionality and the ways of implementing design principles according to the expertise of participants.

Overall, using scenarios as starting points and then analyzing them in terms of elements of the activity theory proved to be effective in learning about less obvious aspects of how the system could be used; the process of the analysis encouraged a long and detailed discussion where the perspectives of regular potential users, technical constraints, and suggestions from experts were discussed at the same time. Involving both experts and potential users allowed a multifaceted discussion and enabled the evaluation of what would be realistic in terms of the design. It enabled the researchers to critically evaluate the proposed ideas, looking at them from more than one person's perspective. This knowledge made the process of deriving requirements from activity theory diagrams much simpler.

**Table 4.** The original design requirement.

| # | Description | Source Design Principle | User Type |
|---|---|---|---|
| DR1 | The system supports three user roles: front-end users, weather-station users, and administrators. | DP1 | All users |
| DR2 | Each role has discrete access/view/write rights in the system. | DP1 | All users |
| DR3 | For different roles, different user interface layouts are supported. | DP1 | All users |
| DR4 | A log-in and log-out mechanism should be supported. | DP1 | All users |
| DR5 | The system should answer questions related to pluviometric data and DEMs of each monitored area. | DP2 | All users |
| DR6 | The weather-station users should be able to update data periodically, either automatically or manually, over their network channel. | DP3 | Weather-station users |
| DR7 | The weather-station users should be able to update data, either automatically or manually, over their network channel. | DP3 | Weather-station users |
| DR8 | The system should provide communication mechanisms (e.g., text-based, messaging, and quick-call actions) to support the communication between weather-station users and local authorities. | DP3 | Weather-station users |
| DR9 | Administrators should be able to have full access to the system. | DP4 | Administrators |
| DR10 | Administrators should be able to maintain and modify the system functions, users and roles. | DP4 | Administrators |
| DR11 | Administrators should ensure the integrity and security of the data provided in the system. | DP4 | Administrators |
| DR12 | Administrators should keep the performance, efficiency, and reliability of the system at the highest level. | DP4 | Administrators |
| DR13 | Front-end users should be able to provide pluviometric information to the system, which should be moderated first. | DP5 | Front-end users |
| DR14 | Front-end users should be able to view historical pluviometric data. | DP5 | Front-end users |

**Table 4.** *Cont.*

| # | Description | Source Design Principle | User Type |
|---|---|---|---|
| DR15 | Front-end users should be able to search for pluviometric data using filtering mechanisms of various conditions. | DP5 | Front-end users |
| DR16 | Front-end users should be able to get information about available samples. | DP5 | Front-end users |
| DR17 | The system should support a communication channel among all user types. | DP6 | All users |
| DR18 | Weather-station users should be notified when a change is made regarding the data they manage. | DP6 | Weather-station users |
| DR19 | Weather-station users should be able to approve or reject system states regarding pluviometric data (e.g., verify data received from front-end users). | DP6 | Weather-station users |
| DR20 | Weather-station users should be able to add users and assign them local weather-station rights. | DP6 | Weather-station users |
| DR21 | Weather-station users should be able to collaborate to manage pluviometric data of common interest. | DP6 | Weather-station users |
| DR22 | Administrators should be notified when new users or data have been added into the system. | DP6 | Administrators |
| DR23 | Administrators should be able to manage users' rights and roles. | DP6 | Administrators |
| DR24 | Administrators should be able to communicate with weather-station users regarding the management of system functions. | DP6 | Administrators |
| DR25 | Front-end users should be able to share information with the community. | DP6 | Front-end users |
| DR26 | Front-end users should be able to notify administrators of system errors. | DP6 | Front-end users |
| DR27 | All functions supported for each user type should be provided via the user panel tool. | DP7 | All users |
| DR28 | The functions provided by the user panel should be implemented through a web interface. | DP7 | All users |
| DR29 | The users should be able to manage their profile through the user panel. | DP7 | All users |
| DR30 | The user panel should allow the weather-station users to upload weather data related to their authorized areas. | DP7 | Weather-station users |
| DR31 | The user panel should provide notification messages to weather-station users when weather data are uploaded by front-end users or other weather-station users. | DP7 | Weather-station users |
| DR32 | The user panel should provide weather-station users with weather data management tools. | DP7 | Weather-station users |
| DR33 | An API should be provided to the weather-station users. | DP7 | Weather-station users |
| DR34 | Through the API, the weather-station users should be able to push and pull information regarding the date, the map, the location (of the weather station), time range, time series analysis, and sophisticated queries regarding a list of events. | DP7 | Weather-station users |
| DR35 | The user panel should provide the administrators with tools to check and fix issues critical to system performance, security, and operation. | DP7 | Administrators |
| DR36 | The user panel should provide accounts settings configuration tools to administrators, providing rich information for each system user (e.g., username, telephone and email address). | DP7 | Administrators |

**Table 4.** *Cont*.

| # | Description | Source Design Principle | User Type |
|---|---|---|---|
| DR37 | The user panel should provide the administrators with tools regarding the back-end functions of the system, such as maps and management tools, supporting functions for inserting, modifying, and deleting system elements. | DP7 | Administrators |
| DR38 | The user panel should provide the front-end users with functions to upload weather data from their area. | DP7 | Front-end users |
| DR39 | The user panel should provide the front-end users with search tools, including advanced search features. | DP7 | Front-end users |
| DR40 | The user panel should notify the front-end users regarding their requests status (e.g., upload new weather data or set a new query). | DP7 | Front-end users |
| DR41 | The users should be able to set requests to the system regarding pluviometric data. | DP8 | |
| DR42 | The system should support different types of pluviometric data, e.g., rainfall level, temperature, and moisture. | DP8 | |
| DR43 | The system should perform the temporal/spatial analysis on the basis of various parameters, such as the pluviometric data variable and geolocation information. | DP8 | |
| DR44 | The users should be able to refine their search, by applying filters regarding the type of the requested pluviometric data, as well as the requested time period and location. | DP8 | All users |
| DR45 | The users should be able to combine different types of queries to produce a sophisticated mixed query, such as pluviometric variable, location, and date (e.g., what is the *maximum level of rainfall* for *November 2015*?). | DP8 | All users |
| DR46 | The users with higher access/write rights (i.e., administrators and weather-station users) should be able to modify (e.g., edit or delete) pluviometric data, applying the aforementioned filters. | DP8 | Administrators/ weather-station users |
| DR47 | The users with higher access/write rights (i.e., administrators and weather-station users) should be able to modify (e.g., add, edit or delete) pluviometric data types and characteristics. | DP8 | Administrators/ weather-station users |
| DR48 | The users should be able to import and export pluviometric data in a common format (e.g., CSV format). | DP8 | All users |
| DR49 | The system should automatically scan for errors on data import action and notify the users accordingly. | DP8 | |
| DR50 | The system should keep a history of the pluviometric data requests. | DP8 | |
| DR51 | The system users should be able to set temporal/spatial analysis queries. | DP9 | |
| DR52 | The system should provide statistical analysis of pluviometric data for both spatial and nonspatial input. | DP9 | |
| DR53 | The administrators and weather-station users should be able to manage the supported locations, including the functions of adding, deleting or updating a location. | DP9 | Administrators/ weather-station users |
| DR54 | The administrators and weather-station users should be able to test and validate the locations on the map and their proper visualization. | DP9 | Administrators/ weather-station users |
| DR55 | The system should be able to interpret the longitude and latitude data into map locations (and vice versa). | DP9 | |
| DR56 | The users should be able to view weather information about any given location (along with any changes made). | DP9 | All users |

**Table 4.** *Cont.*

| # | Description | Source Design Principle | User Type |
|---|---|---|---|
| DR57 | The users should be able to provide and view information about past time periods. | DP10 | All users |
| DR58 | The system should provide the users with historical information about any requested pluviometric data type or request. | DP10 | All users |
| DR59 | Time-series analysis report tools should be used for nonspatial data, providing meaningful historical information to the system users. | DP10 | All users |
| DR60 | Administrators and weather-station users should be able to modify historical data. | DP10 | Administrators/ weather-station users |
| DR61 | The system should provide a detailed view of each location to the users, including information about their types, insert/update date, and coordinates. | DP11 | |
| DR62 | The system should provide a detailed view of each map to the users, including information about their types, insert/update date, boundaries, and covered areas. | DP11 | |
| DR63 | The users should be able to view information related to variables such as rainfall, precipitation, moisture, runoff, and recharge in a specified timeframe. | DP11 | All users |
| DR64 | The system users should define the time period (e.g., day, month, and year) of any request in a graphical way. | DP11 | All users |
| DR65 | The system should provide the users with a graphical option set regarding the values of the requested pluviometric data (e.g., average, minimum, and maximum values). | DP11 | All users |
| DR66 | Various graphical types should be used to visualize the obtained information in the most appropriate way each time (e.g., bar chart and time series lines). | DP11 | |
| DR67 | Interactive markers on the maps should be used to create an enhanced and more efficient interaction experience between the user and the system. | DP11 | |
| DR68 | Color schemes should be supported to visualize the different states of the elevation, slope, aspect, and accuflux parameters on each map. | DP11 | |
| DR69 | Warning message and notifications should be displayed in a meaningful graphical way. | DP11 | All users |
| DR70 | The system should follow a minimal and aesthetic design approach. | DP11 | |
| DR71 | Icons (along with textual information) should be used across the system layout to accelerate users' visual perception. | DP11 | |
| DR72 | Usability heuristics should be applied across the system layout. | DP11 | |
| DR73 | The system information architecture and layout should be adaptive, providing assistive functions attuned to accessibility standards. | DP11 | |
| DR74 | The system should support maintenance and security actions by all authorized administration personnel. | DP12 | |
| DR75 | The system should provide access to authorized system users only to critical system files, such as error logs, security information, and storage mechanisms. | DP12 | Administrators/ weather-station users |
| DR76 | The system should deliver the supported weather data to the weather-station users with high accuracy. | DP12 | Weather-station users |

**Table 4.** *Cont.*

| # | Description | Source Design Principle | User Type |
|---|---|---|---|
| DR77 | The maintenance and installation of data-collection devices should be performed by authorized personnel only. | DP12 | Administrators/ weather-station users |
| DR78 | The weather stations should provide the front-end users with accurate data of high integrity. | DP12 | Weather-station users |
| DR79 | The system should support low-energy consumption schemes. | DP12 | |
| DR80 | The system should be flexible and lightweight to be easily accessed by various devices with poor internet connection. | DP12 | |
| DR81 | The system should support the creation of new user types and roles. | DP12 | Administrators |
| DR82 | The system should provide supportive tools (e.g., a thorough and user-friendly documentation manual) to help its users to overcome problematic situations. | DP12 | All users |
| DR83 | The defined set of rules and regulations should be thoroughly described in a technical report document, providing information about the system functionality and extensibility. | DP13 | Administrators/ weather-station users |
| DR84 | The system should provide testing and validation functions to the administrators regarding the navigation on the maps and the various transformation effects that are applied to them. | DP13 | Administrators |
| DR85 | The system should provide the administrators with tools to define the rules by which a search is performed (e.g., search variables). | DP13 | Administrators |
| DR86 | The rules applied for each search should be tested, validated, and modified by users with high access/write rights (i.e., administrators and weather-station users). | DP13 | Administrators/ weather-station users |
| DR87 | The information architecture of the system should follow common approaches and widely accepted conventions regarding the digital content, metaphors used, and information flow. | DP13 | All users |

## 5. Evaluation Process and Protype App (RQ3)

### 5.1. Qualitative and Quantitative Usability Evaluation

5.1.1. Methodology

The first stage of prototype assessment was qualitative and aimed to discover how users engage with such a system, what is unclear to them, and their expectations and mental models. The mobile application and administration website prototypes were evaluated and deemed a single prototype. This number was determined on the basis of the relatively large number of jobs (to test different components of the prototype) and time and financial constraints. The recruitment process aimed to ensure diversity in terms of age, gender, education level, experience with mobile applications, experience with weather-oriented systems, and familiarity with the GIS field; however, potential participants who lacked experience with mobile applications and weather-oriented systems were not included because they were not part of the target audience. Participants' ages ranged from 21 to 41 (mean: 30), both genders were equally represented, and their levels of education ranged from undergraduate students to PhD holders. Moderated user testing sessions included introducing participants to the study and procedure, having them perform a series of predefined tasks while thinking aloud, and asking them to comment on the user interface, as well as identify the primary benefits and drawbacks of the system at the conclusion of the session. Each participant was required to complete nine activities in 15 min; there were 24 tasks in total, with three people doing each activity. As this study was qualitative and more concerned with observing participants' actions and cognitions than with collecting data, it was decided that a sample size of three people per task was enough. The goal of the test was to allow users to engage with all primary components of the prototype, monitor

their behavior, and identify any problems. The task creation procedure was as follows: first, a use case diagram (Figure 4) depicting all main use cases and tasks for each user role was produced; next, essential activities for participants were defined. Note that certain use cases, such as login into the system, are relevant to many roles. Moreover, obtaining pluviometric data is of utmost relevance to our system's users; as such, several tasks are associated with it (based on the pluviometric variables and attributes, location-based characteristics, date and time periods, etc.). At least one job existed for each design concept.

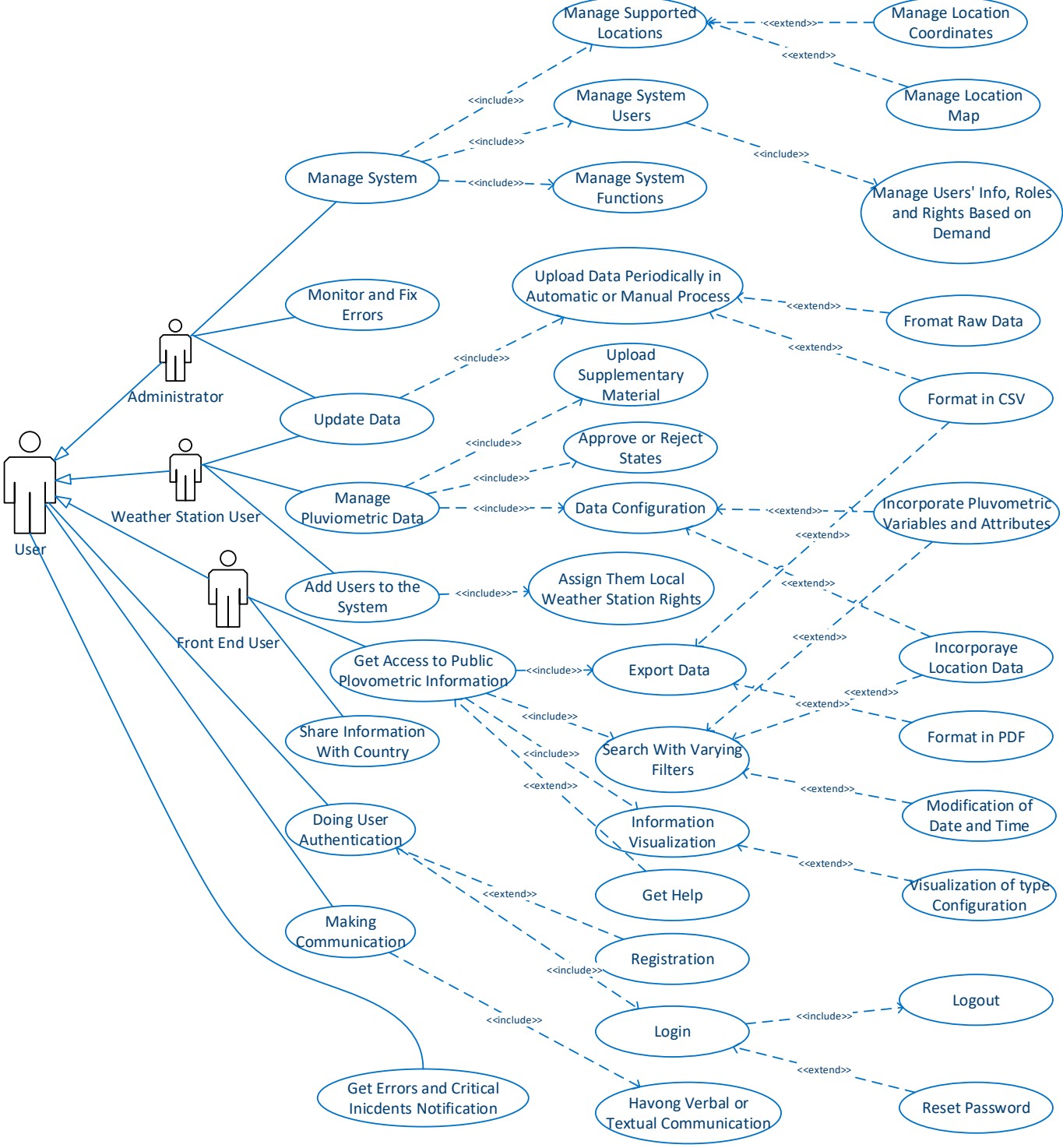

**Figure 4.** Use case diagram.

Figure 4 specifically shows us how the shortcomings of different existing app systems are improved by adopting our approach. We can see there are three core actors. Front end users will aid in collecting and sharing information from a mass data generation point of view. Weather-station users act as formatting and checking unit. Lastly, administrators act as the managing body. Through the incorporation, collaboration, and integration of these three actors, different use cases are defined and tasks are executed.

One of the core goals of this research is to eliminate the shortcomings of different types of existing flood warning apps, as shown in Table 1. In order to do that, we incorporated tasks specifically tailored to three core users as mentioned below. Customization capability, reliability, accuracy, ease of access, and use were some of the main considerations for choosing these targeted user groups for assigning the tasks.

Tasks for administrators (admin website—See Figure 5):

1. Log in to the back-end system.
2. Reset the password of a given system user.
3. Create a new map.
4. Delete an existing map.
5. Edit an existing location by changing its latitude.
6. Enter pluviometric data for a given location.
7. Upload a CSV file of valid pluviometric data for a given location.

Tasks for weather-station users (admin website):

1. Push daily rainfall data to the system.
2. Upload data in raw format.
3. Get the data setting a given query.
4. Log out.

Tasks for front-end users:

1. View the rainfall values in London during November (front-end app).
2. View the daily precipitation during last August.
3. Find the minimum rainfall level, e.g., from 12 October to 19 October.
4. Find the average moisture level in Bristol, e.g., during the summer.
5. Find the time and amount of the highest runoff in the country, e.g., during the summer.
6. Find the number of consecutive days it rained, e.g., in London last year.
7. Find which city of the country had the most rainfall last year.
8. Find when the rainfall levels were in a given range, e.g., between 300 and 400 mm.
9. Show a bar graph of rainfall data, e.g., on 31 December 2014.
10. Estimate which area has the highest level of moisture.
11. Perform a slope image of a DEM.
12. Render the map in greyscale.
13. Get help with the map view.

Some of the duties are relevant to more than one position (such as logging out); however, for the purpose of simplicity, they were allocated to a single role. Communication mechanism design requirements were outside the scope of the high-fidelity prototype. Although participants were unlikely to be administrators of such a system or users of weather stations, they were asked to perform administrative and user tasks to ensure basic usability. The goal was to make the system intuitive even for users with no prior experience, thereby reducing the amount of training required for new users. Pilot research with two participants determined that recording the testing sessions was important; in addition to the recordings, facilitators made notes throughout the procedure and recorded their thoughts afterward to prevent forgetting specifics. The data were subsequently evaluated. The two researchers (who were also facilitators) exchanged and reviewed their notes and observations, separately examined the recordings, and made a note of every difficulty noticed or stated by participants for each activity. They searched for pain spots and places for growth, as well as the pathways individuals followed to attain their objectives,

unexpected behaviors, etc. At the conclusion of this step, the two researchers examined their analytical results and merged their suggestions.

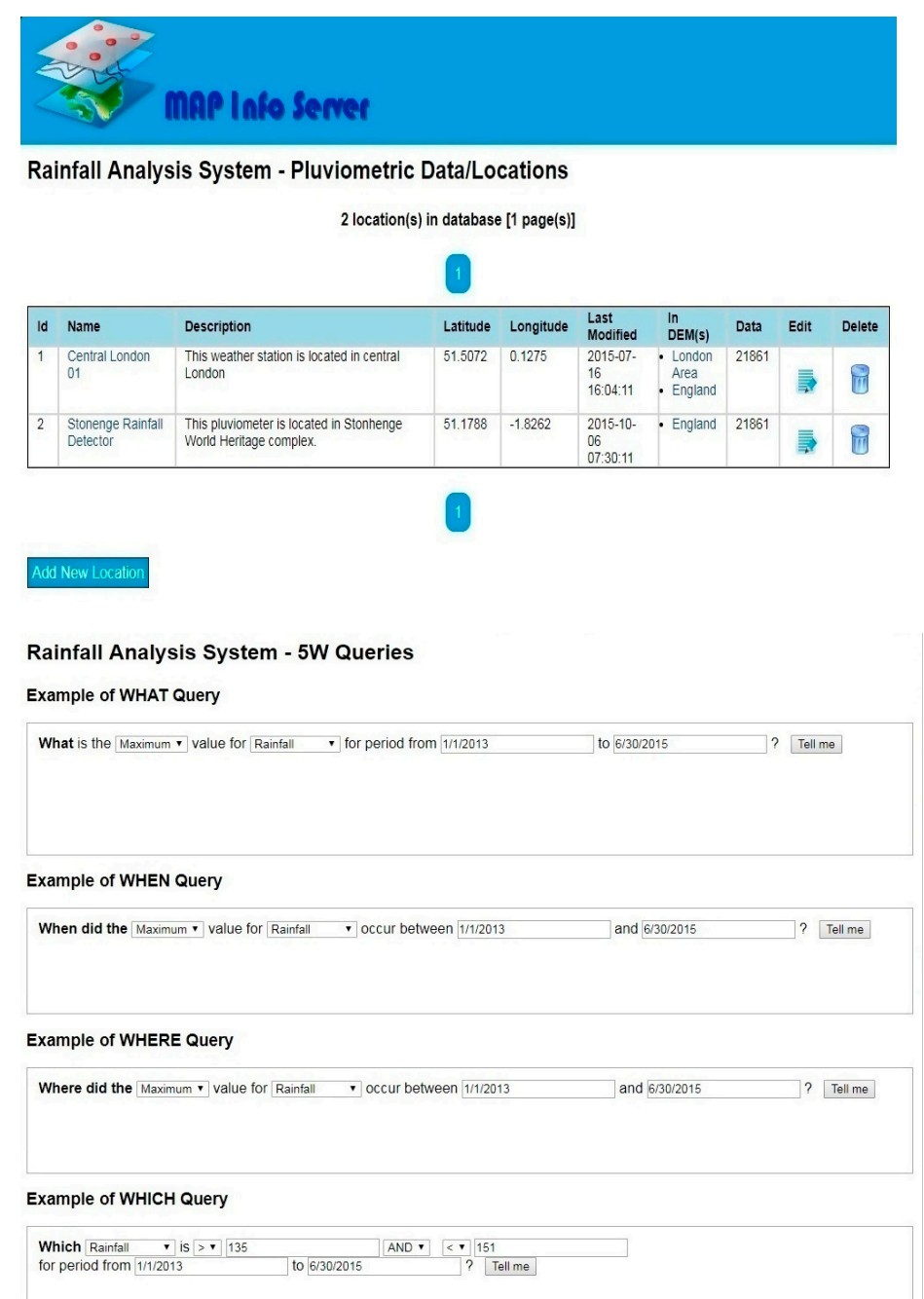

**Figure 5.** Some of the designed prototype screens for admins and weather-station users.

5.1.2. Outcome

The mobile application interface was clear enough for participants to be able to complete most of the predefined tasks; however, not everything matched their expectations or was easy to understand, such as the distinction between statistical and graphical information. Viewing information was often frustrating, especially when participants wanted to make a quick change (e.g., to update the displayed information); they explicitly stated that they wanted to be able to update the displayed information without having to step back and restart the process. Another issue raised during the sessions was the possibility of viewing combined information, such as answers to "what" and "when" questions. The

current design does not allow this functionality; however, participants stated that it would be useful to be able to answer more than one question at the same time. When asked about the update action, the participants stated that they would prefer the update to be automatic, but they wanted to be able to see when the data of the system were last updated. Another issue was that participants were confused about the location of the statistical and graphical data provided since the prototype did not provide any information about the related locations or the available weather stations. The updated prototype should address these issues.

Regarding the administrator interface, the findings were unambiguous. The participants wanted quick access to all the main activities, and the horizontal menu was not convenient for this; they preferred it on the right side of the screen, so that they could control it easily when using a mobile device (e.g., a tablet), as most of the users were right-handed. The identified issues and their impact are presented in Table 5.

**Table 5.** Identified Issues.

| Screen | Description | Impact | Recommendation |
|---|---|---|---|
| All screens | The horizontal menu at the top of the screen was inconvenient. | The horizontal menu slowed down users' automated processes due to having to scan the whole width of the screen. | Provide the menu vertically to the right to allow quick access to admin panel menu items. |
| Statistics screen | The prototype does not support combined information seeking. | Users must start a new query to answer different questions for the same data. | Ensure the prototype allows for combining queries. |
| Landing screen | Users were confused by the statistical and graphical type options. | Users were confused about spatial and nonspatial data and unsure of the meaning of graphical and statistical types. | Ensure there is a clear distinction between the statistical and graphical information. |
| Graphical analysis screen | The prototype does not allow for quickly making modifications to the presented results. | Users were annoyed with having to go to the previous screens to make small modifications to the presented data. | Provide a quick and easy way to modify the search on the same screen. |
| Statistics initialization screen | The prototype does not support combined information seeking. | Users must start a new query to answer different questions for the same data. | Ensure the prototype allows for combining queries. |
| All screens | Update must be executed manually by the user. | This functionality adds a secondary task for the user. As such, it may be forgotten or overlooked, which would mean outdated data were presented. | Ensure any required updates are executed automatically. The users should be informed as to when the last update took place. |
| No specific current screen | The prototype does not provide any information on the location of the presented data. | The user is unaware of which location the presented data refer to. | Present location-based information related to the data. |
| No specific current screen | The prototype does not provide any information on the location of the weather stations. | The user is unable to check whether there is a weather station near a location. | Provide a map with all the available weather stations. |

The researchers subsequently reflected on the design principles and requirements: whether they were extensive enough to cover all user needs and whether some requirements were less relevant since users prefer carrying out tasks in another way. Although existing design requirements appeared relevant, they were found not to cover everything; hence,

the list of design requirements was extended by adding six new requirements: a vertical menu for administrators; an ability to make sophisticated queries combining different types of information; updating the system data automatically and showing the date of the last update; showing weather station-related information, such as location on a map and available samples; the easy distinction between spatial and nonspatial data; easy-to-access and independently-supported queries. Table 6 lists the newly added design requirements.

**Table 6.** Additional design requirements discovered by user testing.

| # | Description | User Type | Issue Addressed |
|---|-------------|-----------|-----------------|
| DR88 | The menu should be displayed vertically to the right side of the screen. | Administrator | A1 |
| DR89 | Users should be able to make sophisticated queries combining different types of information. The queries are normally stated as what, which, and where types and can be applied to all or a selection of the data stored in the system. | Weather station, Front-end users | W1, F3 |
| DR90 | The system data should be updated automatically, and the user should be able to view when they were last updated. | Front-end users | F4 |
| DR91 | Users should be able to view weather station-related information, such as location on a map and available samples. | Front-end users | F6 |
| DR92 | Spatial and nonspatial data should be easily distinguishable by users. | Front-end users | F1 |
| DR93 | Queries should be supported independently and should be easy to access. | Front-end users | |

The prototype was updated accordingly. A new information architecture approach was implemented to enable users to quickly distinguish between the two major types of data provided by the system while providing quick access to a weather station's location, as well as sampling data and information about the currently selected location. A map view of the areas with available data was considered a valuable add-on to the interface. Lastly, to ensure the system is expandable and to satisfy the users' need to view their current location on a map, an LBS (location-based service) section was added. Figure 6 shows some of the updated app screens. One change was made to the administrator panel; a vertical menu was added (Figure 7).

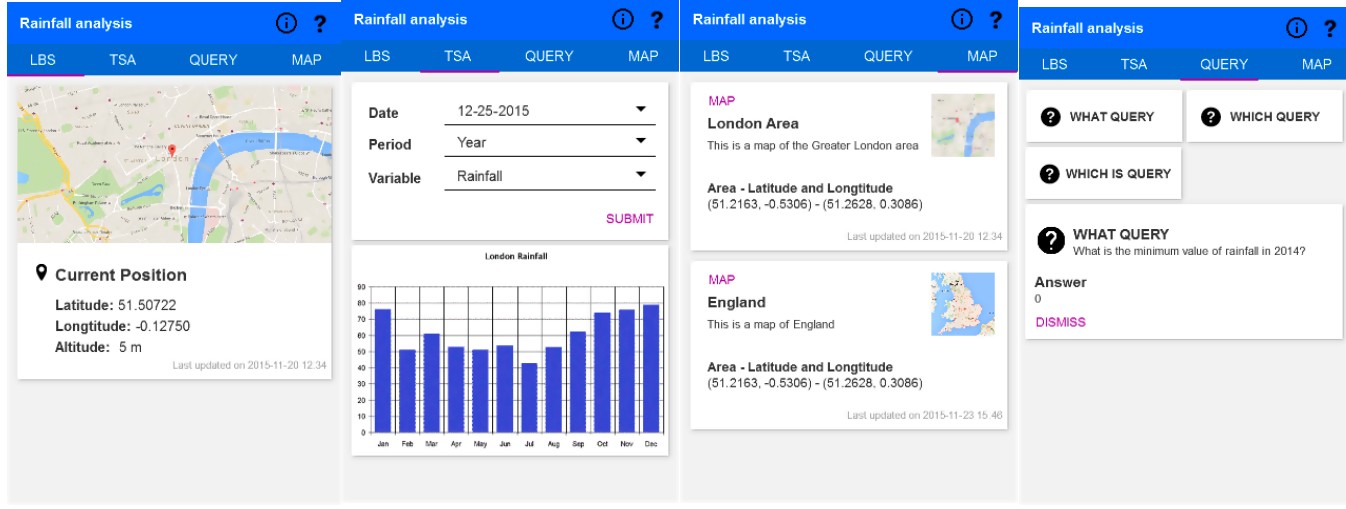

**Figure 6.** Some of the updated prototype screens for front-end users.

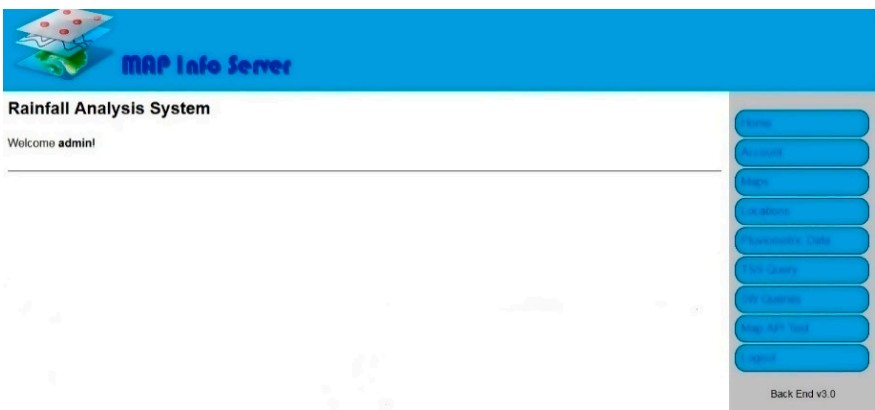

**Figure 7.** The newly added menu for admins.

*5.2. Quantitative Usability Evaluation*

5.2.1. Methodology

After refining the prototype, there was a quantitative round of user testing to check whether various usability metrics were within the expected range and to obtain some benchmark figures for other potential rounds of testing. The reason for focusing on quantitative data was that, while the first round of testing provided a decent amount of qualitative data to understand the underlying reasons, there was not enough data to quantify attitudes and generalize the results. Some qualitative data were also collected to understand the observed behavior and address the discovered problems more effectively. User testing (including collecting usability metrics and observing participants for unexpected behavior), questionnaires, and interviews were all used.

Fifty-eight participants were recruited for the study; as in the previous round, variation in terms of demographics and levels of relevant experience (excluding people completely inexperienced with mobile apps and weather systems) was important in selecting participants. The ages of participants ranged between 19 and 67 years (with the mean age distribution being 32.9 years); 48% of the participants were male, and 52% were female; in terms of educational and professional level, they comprised undergraduate students, postgraduate students, PhD holders, professionals, and retired professionals.

After being introduced to the purpose of the study and the system being tested, participants were asked to carry out seven tasks in 15 min; the allocated time was based on an estimate of 2 min to complete a task (based on researchers' estimates and confirmed by a pilot study), plus an extra minute to allow for switching between very different tasks. The tasks were fewer in number than in the previous testing round, focusing on the front-end users only (the front-end user interface appeared to be the most problematic in the previous testing stage).

The tasks:

1. View the average rainfall during November.
2. Find the location with the maximum available samples.
3. Find the location with the lowest precipitation between September and November.
4. Perform an accuflux of an existing DEM.
5. Find which areas had rainfall between 20 and 23 mm during the summer.
6. Estimate which area had the highest moisture levels on 15 November.
7. Display a rainfall graph for the month of January 2012.

All tasks represent the user goal of viewing pluviometric data, which is the main function of the system and one of the most important aspects of system design to get correct. This particular aspect of functionality is going to be used the most, and any issues that arise in this area are likely to be the least tolerated by potential users as it is the core functional component of the app and establishes the most notable part of the user's use-case experience.

The next step was a questionnaire with 21 questions about their experience with the application; participants were asked to specify a number from one to five on a Likert-type scale, with one being "strongly disagree", and five beings "strongly agree". The questionnaire items were based on the USE method proposed by Lund [44], which is a credible usability measuring tool [45–47]. It focuses on the subjective experience with the system, which supplements usability metrics, revealing not only how well users performed but also how they felt about their performance and their interaction with the system.

The USE questionnaire:

Usefulness (Questions 1–6):

- The application helps me be more effective.
- The application helps me be more productive.
- The application is useful.
- The application makes the things I want to accomplish easier to achieve.
- The application saves me time when I use it.
- The application does everything I would expect it to do.

Ease of use (Questions 7–13):

- Using the application is easy.
- Using the application is simple.
- The application is user-friendly.
- Using the application is effortless.
- I can use the application effectively without any written instructions.
- I do not notice any inconsistencies in using the application.
- I can use the application successfully every time.

Ease of learning (Questions 14–16):

- I quickly learned how to use the application.
- I easily remember how to use the application.
- I easily learn how to use the application.

Satisfaction (Questions 17–21):

- I feel satisfied with what the application provides.
- I would happily recommend the application to a friend or colleague.
- The application is fun to use.
- The application is pleasant to use.
- The application works as expected.

One-to-one interviews with each participant followed, focusing primarily on their experiences with the application. They were also asked their opinions on the design used. During this process, facilitators tried to clarify any questions they had regarding the participants' behavior during the testing session or the comments they made. Rich insights were gained, as the participants had a more holistic and clearer picture of the system after the task completion and questionnaire.

### 5.2.2. Outcome

Various usability measures were computed after the testing session, including effectiveness (accuracy and completeness of user goal attainment), efficiency (time spent), and satisfaction. The recordings of interviews and testing sessions, in addition to the facilitators' notes, were thoroughly analyzed to determine the rationale behind the measures.

In terms of design, the system quality and satisfaction questionnaire revealed that participants generally enjoyed using the application (the average assessment for "The application is pleasant to use" was 4 out of 5, and that for "The application is entertaining to use" was 3.9 out of 5). This was supported by the outcomes of the interviews, in which the majority of users (77.5%) indicated that the design of the mobile application was intuitive and visually pleasing. Participants were able to recollect the steps they took to perform

a job with excellent accuracy, most likely because the user interface adhered to criteria provided by the device maker. Participants neither saw nor acknowledged any distractions.

In terms of functionality and usability, the task completion rates and times, as well as the surveys and interviews, suggested that the system was properly operating.

In terms of effectiveness, the average completion percentage of all activities (total system effectiveness) was quite high (91.38%), particularly given that it is not a mission-critical system, participants were using it for the first time, and not all participants had extensive prior knowledge. The 95% confidence interval for the total completion rate was 88% to 98.7%, indicating that, on average, each individual would complete all activities with a success percentage ranging from 88% to 98.7%. Despite their difficulty, the most difficult tasks (tasks 2, 4, and 7) showed high completion rates (89.66%, 87.93%, and 87.39%),greater than the average task completion rate (78%), according to many studies [48,49]. These tasks were deemed the most challenging by research participants and had the highest mistake rates. Although these areas need more attention in the long term, the completion rates were acceptable; consequently, they do not require immediate modification but should be the subject of a subsequent examination.

The proportion of participants who committed one or more mistakes while completing each task varied from 51.7% (Task 5) to 74.1% (Tasks 2 and 4); Table 7 displays these percentages for each task.

**Table 7.** Percentages for each task.

|  | Task 1 | Task 2 | Task 3 | Task 4 | Task 5 | Task 6 | Task 7 |
|---|---|---|---|---|---|---|---|
| Proportion who committed 1 or more errors | 67.2% | 74.1% | 58.6% | 74.1% | 51.7% | 63.8% | 72.4% |
| 95% upper limit | 79% | 85% | 71% | 85% | 64% | 76% | 84% |
| 95% lower limit | 55% | 62% | 46% | 63% | 38% | 51% | 61% |
| Expected proportion according to Sauro's analysis | 67% |  |  |  |  |  |  |
| $p$-value (chi-square) | $p > 0.05$ | $p > 0.05$ | $p > 0.05$ | $p > 0.05$ | $p > 0.05$ | $p > 0.05$ | $p > 0.05$ |

In Table 8, the aforementioned efficiency metrics for each activity are shown.

**Table 8.** Summary of the efficiency metrics.

|  | Task 1 | Task 2 | Task 3 | Task 4 | Task 5 | Task 6 | Task 7 |
|---|---|---|---|---|---|---|---|
| Time $_p$ (task completion time, i.e., average time to complete the task in seconds) | 92.46 ± 25.49 | 92.44 ± 28.29 | 77.33 ± 24.93 | 105.90 ± 13.65 | 80.56 ± 13.45 | 99.02 ± 16.08 | 100.30 ± 15.78 |
| Time $_E$ | 44.09 | 27.51 | 45.48 | 26.4 | 47.86 | 42.74 | 32.71 |
| Standardized time (difference between the actual time and expected time/standard deviation) | −1.08 | −0.97 | −1.71 | −1.03 | −2.9 | −1.3 | −1.2 |
| Quality level | 86% | 83.5% | 95.6% | 84.9% | 99.8% | 90.4% | 89.4% |
| $\bar{P}$ (relative time-based efficiency) | 87.40% | 81.65% | 85.29% | 81.08% | 89.14% | 88.13% | 80.24% |
| $\bar{P}_E$ (relative expert time-based efficiency) | 41.67% | 24.30% | 50.16% | 20.22% | 52.96% | 38.04% | 26.16% |

User happiness is another measure of strong functioning and usefulness. The analysis of the questionnaire responses (Table 9) indicates that the application was perceived as useful; participants tended to agree that the app was consistent in design and provided the features that they required during the course of the study, and overall satisfaction was high. Numerous past studies [49,50] have indicated that the mean score on a scale from 1 to 5

for systems with acceptable usability is 4. When this was explored in the present research, all assessed dimensions received a mean score of around 4 and an overall score of 4. This gives an additional assessment of the app's quality.

**Table 9.** Summary of use questionnaire results.

| Criterion | Usefulness | Ease of Use | Ease of Learning | Satisfaction | Overall |
|---|---|---|---|---|---|
| Mean score | 3.94 | 3.93 | 4.08 | 4.02 | 4 |
| SD (Standard deviation) | 0.34 | 0.33 | 0.5 | 0.41 | 0.23 |
| Lower 95% CI (confidence interval) | 3.86 | 3.83 | 3.95 | 3.9 | 3.93 |
| Upper 95% CI | 4 | 4 | 4.2 | 4.12 | 4.05 |

The visual components (e.g., icons and maps) were quite familiar to the participants, who reported having a clear understanding of the material.

The average time-based user efficiency was 0.69 tasks per minute, indicating that users were able to accomplish 70% of each job within 1 min. This was a great outcome, given that the activities involved viewing many screens and active interaction.

As measured by the ratio of effective participants' worktime to total participants' worktime, the relative time-based efficiency was, on average, 84.7%. None of the jobs had a time-based relative efficiency below 80%. High relative efficiency (as in this research) implies that users had no difficulty completing any of the activities [51,52], indicating that the software is highly usable.

Comparing participant performance to that of experts (relative time-based expert efficiency) revealed that certain activities may have been more challenging than others. We utilized the Keystroke-Level Model for Advanced Mobile Phone Interaction framework [53], which was based on the ground-truth Keystroke Level Modeling (KLM) framework [54], to estimate the task time for experienced users who made no errors. Activities 2, 4, and 7 were more difficult for novice users to complete than other tasks, which explains why these tasks had lower success rates and greater mistake rates (although still within the expected range).

Observing and questioning participants revealed two more usability issues: users considered the assistance information unsatisfactory, and there was no progress indication, making it unclear if the system is operating when it takes longer to load information. The final fully working prototype will need to fix these shortcomings.

Encouragingly, participant observation and interviews revealed that the flaws noted in the previous study phase had been resolved; none of the participants mentioned comparable concerns when providing comments on the system.

The visual components (e.g., icons and maps) were quite familiar to the participants, who reported having a clear understanding of the material.

Overall, all findings fell within the expected range, confirming that the level of usability was as anticipated; consequently, the prototype as it stands could be transformed into a fully functional system (including solutions to the identified usability issues), and additional refinements could be made later in the application's lifecycle. The qualitative insights gathered through watching and interviewing participants enhanced the team members' comprehension of the users' perspective and the areas that would need further attention in the future.

### 5.3. The Final Prototype

Using dynamic layouts, the final prototype was designed to be useable on a variety of device types, from mobile phones to tablets, with displays ranging from 4 to 10 inches and in landscape and portrait screen orientations (all graphical components automatically rescale on the screen). By using Android support libraries, the user interface (including appearance) can be maintained across all devices.

The primary interface consists of four distinct tabs. Some screens provide a more detailed view of the data to be accessed. Figures 8 and 9 display the application's structure.

To illustrate the functionality of each page, Figure 10 represents the LBS tab for location-based services. It displays customized services based on the user's current location; the location is automatically updated every 10 to 60 s, depending on the device's speed.

The primary interface consists of four distinct tabs. Some screens provide a more detailed view of the data to be accessed.

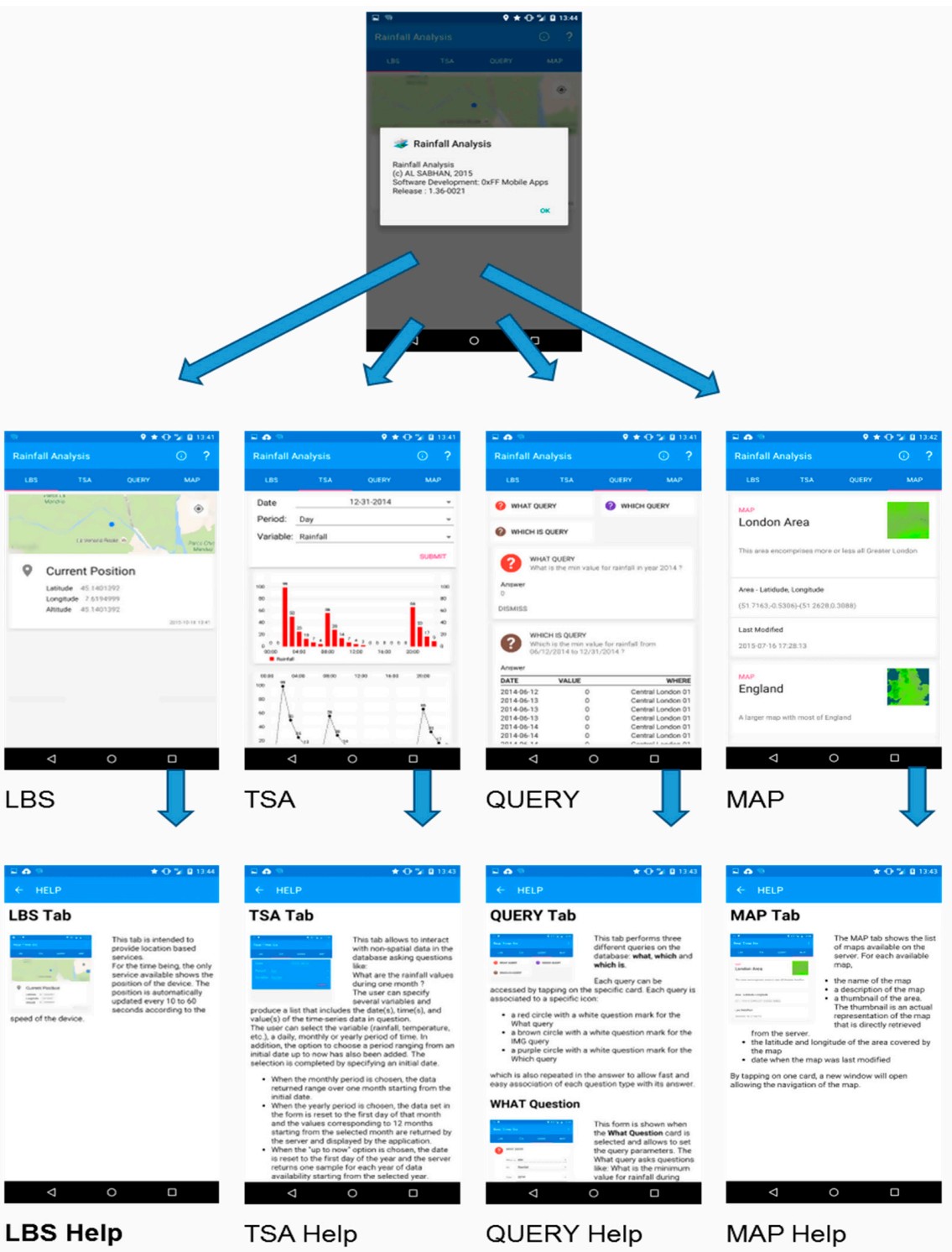

**Figure 8.** The four tabs of the main interface.

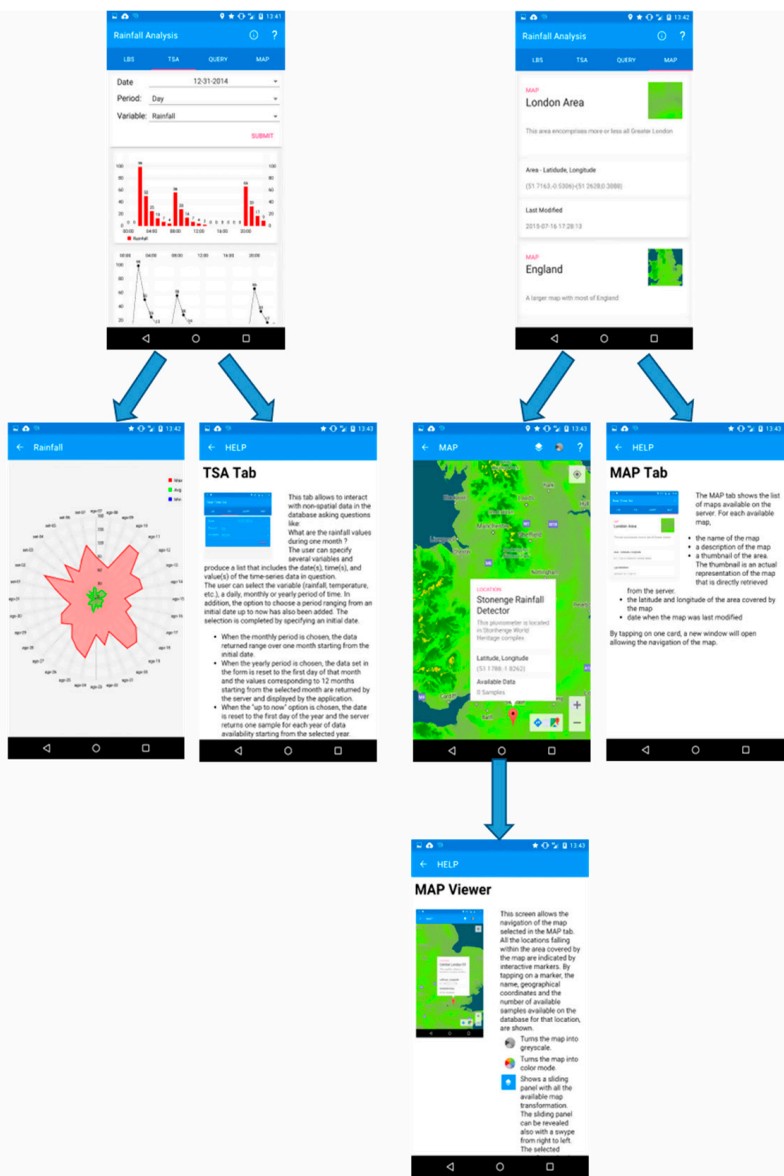

**Figure 9.** A detailed view of the data.

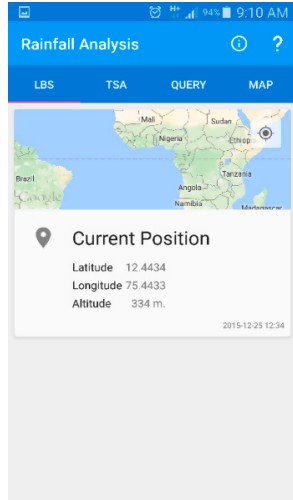

**Figure 10.** The LBS interface.

Figure 11 shows the time series analysis interface and Figure 12 shows the query tab. Figure 12 permits the choice of a query (what, when, where, or which), a function (min or max), and a variable (rainfall, precipitation, soil moisture, runoff, etc.). The construction of a query to meet specific requirements. "What is the lowest soil moisture value during the supplied time period?" is an example of a question that may be processed.

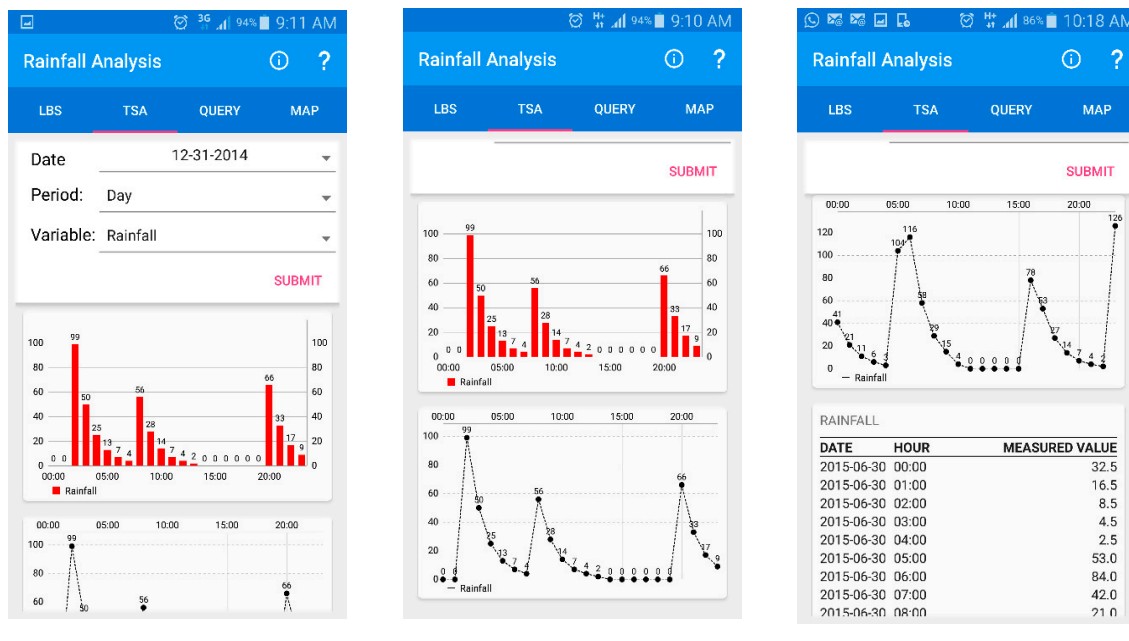

**Figure 11.** The TSA interface.

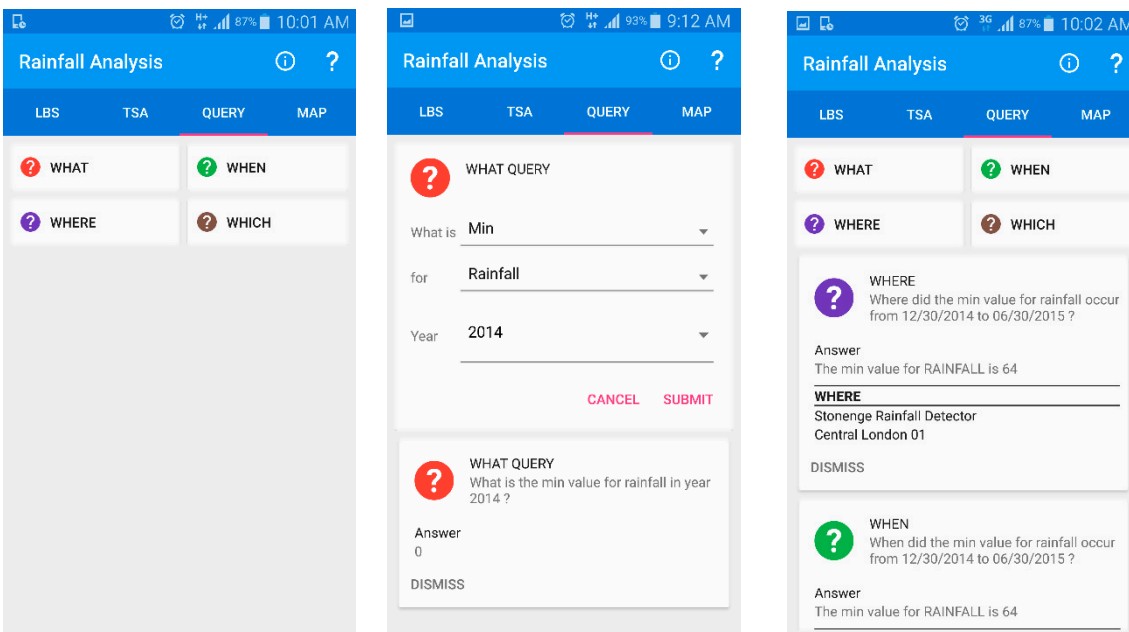

**Figure 12.** The query tab.

The map tab (Figure 13) displays the server maps that are accessible. For each map, the application displays its name and description, a thumbnail of the recognized region (directly downloaded from the server), the latitude and longitude of the area covered by the map, and the modification date.

The map tab provides access to the map viewer. The sliding panel may be uncovered by swiping from the right to the left. This enables the user to perform map analysis,

including slope, aspect, and accuflux. Thus, fundamental geographical markers are shown and may be used as references when scrolling.

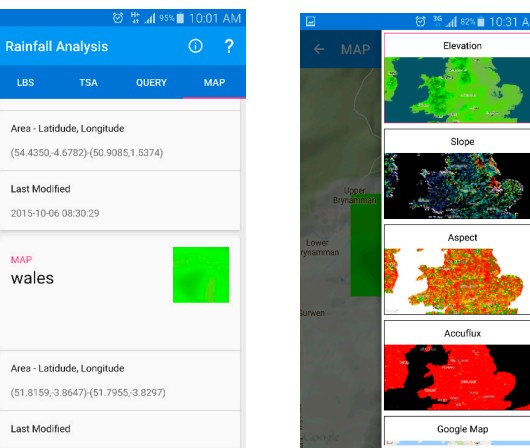

**Figure 13.** The map tab interface and the map viewer.

Due to the well-defined design concepts and needs, as well as the insightful insights obtained throughout brainstorming and user testing sessions, developing the final proto-type was quite uncomplicated. Throughout the design process, this sort of information helped address issues about functionality and design decisions. Sharing this information with the team helped eliminate speculation and enabled choices to be based on reliable data.

## 6. Conclusions and Future Research

Due to the rising frequency and severity of flooding occurrences across the globe, smartphone applications for flood warning have gained significance in recent years. These applications provide several advantages and benefits over conventional flood warning systems. Initially, flood warning mobile applications give users with real-time and accurate information about flooding occurrences in their region, including water levels, predictions, and alarms. This information may assist users in making educated choices on how to prepare for and react to floods, as well as in avoiding potentially hazardous circumstances. Additionally, they can be downloaded and used on a number of mobile devices, such as smartphones and tablets; flood warning mobile applications are generally more accessible than conventional flood warning systems. This implies that consumers may get flood alerts and updates regardless of their location and can obtain vital information quickly and easily during an emergency. By providing users with tools and resources to prepare for and react to floods, mobile applications for flood warning may bolster community resilience. Many flood warning applications, for instance, include checklists and advice for preparing emergency kits, evacuation plans, and other actions to mitigate the effects of floods. Moreover, flood warning mobile applications may lower the related expenses and hazards of flooding. By providing users with real-time information and notifications, these applications may aid in minimizing property damage and saving lives. In addition, by encouraging users to take proactive measures to prepare for floods, these applications may lower the expenses associated with disaster response and recovery activities.

This article provided a revolutionary way to building mobile applications for flood warning that includes concepts of human–computer interaction for best results. The study underlined the need of gaining an early awareness of the users and their jobs in order to properly satisfy their requirements and expectations. Consequently, data gathering processes and system components were essential to this strategy. The primary goal of the research was to gain valuable insights into how potential users would interact with flood warning applications, what technical constraints should be considered during application design, and how to translate these insights into actionable starting points for developing and testing the application. To attain these goals, we analyzed the app's design needs

using expert reviews and scenario evaluations. The research identified three primary user groups for the mobile app for flood warning: front-end users, weather-station users, and administrators. The app's design specifications were determined by the demands and mental models of each user group. For example, the application should offer distinct access, view, and write permissions for each user role, as well as separate user interface layouts for each role. In addition, a log-in and log-out mechanism must be given. On the basis of the knowledge of the participants, a total of 93 design needs were established. To evaluate the efficacy of the app, we employed an evaluation procedure that measured the efficiency of completing tasks, such as identifying the location with the most available samples, calculating the accuflux of an existing DEM, and displaying a rainfall graph for January 2012. According to the assessment findings, tasks 2, 4, and 7 were the most difficult. In addition, we performed an assessment of user satisfaction on the basis of the findings of a questionnaire that rated the usefulness, convenience of use, simplicity of learning, contentment, and overall experience. The mean score for each category was 3.94, followed by 3.93, 4.08, 4.02, and 4.

The research gave useful insights into the design and usability evaluation of a mobile app for flood warning. However, it is important to note the study's possible weaknesses. Firstly, the sample size was rather small, which may restrict the generalizability of the results. Secondly, the research was conducted in a particular area, which may restrict the application of the results to other places. Lastly, machine learning and artificial intelligence methods, which might improve the accuracy of flood prediction models, were not fully incorporated into the research.

As for suggestions for future work, the accuracy of the flood prediction models might be improved by using more powerful machine learning and artificial intelligence approaches. In addition, user testing with a bigger and more varied sample size might increase the generalizability of the results. Moreover, it would be advantageous to compare the efficacy of the mobile app for flood warning to that of other current flood prediction and warning systems. Expanding the research to include more geographic places with varying climatic circumstances might offer a more complete understanding of the design and use of flood warning mobile applications.

**Author Contributions:** Conceptualization, W.A. and B.D.; methodology, W.A. and B.D.; software, W.A. and B.D.; validation, W.A. and B.D.; formal analysis, W.A. and B.D.; investigation, B.D.; resources, W.A. and B.D.; data curation, W.A. and B.D.; writing—original draft preparation, W.A. and B.D.; writing—review and editing, W.A. and B.D.; visualization, W.A.; supervision, W.A.; project administration, W.A. and B.D.; funding acquisition, W.A. All authors have read and agreed to the published version of the manuscript.

**Funding:** This research received no external funding.

**Data Availability Statement:** Data are available on suitable demand.

**Acknowledgments:** The authors would like to thank the laboratory technicians for providing assistance in specimen assembling and testing. The fresh and hardened concrete analysis performed in this paper was carried out at the laboratory of Alfaisal University, Riyadh, Saudi Arabia. The authors would like to thank all the people who participated and helped in the testing of the application, particularly the students, lab technicians, and staff, for their useful contributions in the experimental works. They helped make this research possible.

**Conflicts of Interest:** The authors declare no conflict of interest.

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
