# Peer review of "Real-Time Flood Forecasting and Warning: A Comprehensive Approach toward HCI-Centric Mobile App Development"

_mti, doi:10.3390/mti7050044_

Round 1

Reviewer 1 Report

Authors should summarize the article's main findings and indicate main conclusions and future research. It is important to point out the advantages of flood warning mobile app vs. dashboard user interface (tablet). Also, a comprehensive overview of "comparable systems being difficult to use" and pros and cons of other flood apps on the market. How is your mobile app different?

Author Response

Thank you for your valuable suggestions and feedback. I appreciate the points you made. I have updated the article based on the comments. You will see that section 2 has been updated answering your questions. A new section titled, “Conclusion and Future Recommendations” has also been added.

Please see the responses in the attached doc

Reviewer 2 Report

The work presented in the manuscript is interesting. Overall, the manuscript is well written. However, while reading the manuscript, several concerns were raised that needed to be discussed.

In section C the authors provided a list of design principles, which are repeated in the table following the very same list in Table 2. There is no need to present these design principles with the exact text two times, only in different forms. Instead, I suggest the authors explain how these design principles were constructed. The authors have mentioned that the proposed principles differ from well-known heuristic guidelines such as those from Nielsen and others. It would be good to clarify why they didn't follow these well-known principles and focused on functionality instead of usability. 

Section 5. A is entitled "Quantitative Usability evaluation", although the first statement and some later statements explain the quality evaluation procedures. 

I suggest revising and improving the use case diagram in Figure 4. For example, a "generalization" should be used between different actors, not as a straight line, since a straight line is used for drawing the connection between an actor and a use case. There are several use cases that are connected with other use cases using the <<include>> connection. For example, the use case "Manage system users" is connected with the connection <<include>> with the following use cases: 1) "Manage users' info", 2) "Manage user's roles", and 3) "Manage users' rights". According to the current model, if the administrator runs the functionality for managing a user, each time, the administrator must manage information about the user, the user's roles and the user's rights. This is probably not the case. Overall, the use case diagram shouldn't be used for explaining the logical order of specific tasks or functionalities represented by use cases. The logical order of execution is defined either in the specification of use cases or by modelling a suitable diagram. The names of the use case should also be improved. Use case names should begin with a verb (a use case models an action, so the name should begin with a verb), and must be descriptive. For example, the use case "Location" is not explaining what is meant by the location.

Several tasks are listed in the text that follows Figure 4. How were these tasks prepared? How are these tasks mapped to the functionality provided in the use case diagram in Figure 4?

After the tasks list, there is a screen that shows the interface of the proposed solution. But the figure needs to be explained in the text. The authors could have explained the examples of user forms used to finish the tasks listed above in Figure 5.

In section A-II Outcome the authors start with the statement that the mobile application interface was clear enough for participants. But in the screens provided in Figure 5, no mobile application interfaces are presented. Giving the prototype screens for the mobile application would be better for the reader to understand how the participants tested the proposed solution.

The manuscript must be properly concluded, and the main results discussed. A section that would comprehensively discuss the study's results needs to be included. In the discussion section, the authors should present the main contribution of their work in relation to existing literature. Authors must also identify all the threats to internal and external validity. All limitations must be identified and discussed as well.

Minor observations:

The statement that starts in Line 173 is not completed correctly.

The paragraph that starts in Line 312 repeats the text provided in the previous paragraph (the one that begins in Line 306).

Author Response

Thank you for your valuable suggestions and feedback. I appreciate the points you made. I have updated the article based on it. Irrelevant text related to design principles have been omitted in the revised manuscript.

Regarding not following well-known heuristic guidelines such as those from Nielsen and others:

Heuristic recommendations are often focused on usability criteria such as learnability, efficiency, and mistake avoidance, but they may not include other crucial components of the user experience, including engagement, contentment, and emotional reactions. Additionally, relying only on heuristic rules may result in a limited perspective of usability and may cause designers to overlook other essential information sources, such as user research and user feedback.

Please see the attached responses.

Reviewer 3 Report

In the paper, the authors proposed a real-time flood forecasting and warning system which combines a Geographic Information System (GIS) with dynamic hydrological modelling with a focus on the user experience side of the end product. They analysed and addressed the current user needs and requirements for building a user interface for mobile real-time flood forecasting. The application presented could be a solution with possibilities to be implemented. However, the following aspects should be considered to improve the structure and quality of paper:

1.       In the Introduction, the authors present various approaches from the literature. I think that a synthesis of the solution proposed in the literature depending on the type of analysis is useful for readers. This synthesis can be given as a table.

2.       Avoid the reference loops and a presentation of each reference should be introduced.

3.       The strong points of the proposed approach compared with the other solutions from the literature should be better highlighted.

4.       The introductions of the methodology should be refined with a mathematical tool well done regarding the data processing and forecasting methods.

5.       Section 5 should present the results in another manner, associated with a research paper. The authors should define more analyse scenarios which to differentiate the solutions.  In addition, the authors also have to provide some insightful discussion of the results compared with other systems from the literature

6.       The limits of the proposed application are missing.

7.       The conclusions are missing. Also, future work should be presented.

Author Response

Please see attached responses

Round 2

Reviewer 1 Report

The paper has been improved in accordance with suggestions

Reviewer 2 Report

The authors revised the manuscript with new explanations and content as suggested by the comments given by reviewers. The manuscript is ready to be accepted for publishing.

Reviewer 3 Report

The authors performed changes to the initial manuscript. New explanations, elaborations of details, and revisions have been added. I have no other observations.